# Risk factors for diarrheagenic *Escherichia coli* infection in children aged 6–24 months in peri-urban community, Nairobi, Kenya

**Noah O. Okumu**[1,2]*, **Joseph J. N. Ngeranwa**[2], **Dishon M. Muloi**[1,3], **Linnet Ochien'g**[1], **Arshnee Moodley**[1,4], **Christine Mutisya**[1], **Alice Kiarie**[1], **Joseph O. Wasonga**[1], **Julie Watson**[5], **Maud Akissi Amon-Tanoh**[5], **Oliver Cumming**[5], **Elizabeth A. J. Cook**[1]*

**1** Animal and Human Health Department, International Livestock Research Institute, Nairobi, Kenya, **2** Department of Biochemistry, Biotechnology and Microbiology, Kenyatta University, Nairobi, Kenya, **3** Institute of Infection, Veterinary and Ecological Sciences, University of Liverpool, Liverpool, United Kingdom, **4** Department of Veterinary and Animal Sciences, University of Copenhagen, Frederiksberg C, Denmark, **5** Department of Disease Control, London School of Hygiene and Tropical Medicine, London, United Kingdom

* N.Okumu@cgiar.org (NOO); E.cook@cgiar.org (EAJC)

**Data Availability Statement:** Data is contained within the article.

## Abstract

*Escherichia coli* commonly inhabits the gut of humans and animals as part of their microbiota. Though mostly innocuous, some strains have virulence markers that make them pathogenic. This paper presents results of a cross-sectional epidemiological study examining prevalence of diarrheagenic *E. coli* (DEC) pathotypes in stool samples of asymptomatic healthy children (n = 540) in Dagoretti South subcounty, Nairobi, Kenya. *E. coli* was cultured and pathotyped using PCR to target specific virulence markers associated with Shiga-toxin, enteropathogenic, enterotoxigenic, enteroaggregative, entero-invasive and diffusely adherent *E. coli*. Overall prevalence of DEC pathotypes was 20.9% (113/540) with enteropathogenic *E. coli* being the most prevalent (34.1%), followed by enteroaggregative *E. coli* (23.5%) and Shiga-toxin producing *E. coli* (22.0%) among positive samples. We found evidence of co-infection with multiple pathotypes in 15% of the positive samples. Our models indicated that at the household level, carriage of DEC pathotypes in children was associated with age group [12–18 months] (OR 1.78; 95%CI 1.03–3.07; p = 0.04), eating matoke (mashed bananas) (OR 2.32; 95%CI 1.44–3.73; p = 0.001) and pulses/legumes (OR 1.74; 95%CI 1.01–2.99; p = 0.046) while livestock ownership or contact showed no significant association with DEC carriage (p>0.05). Our findings revealed significant prevalence of pathogenic DEC circulating among presumptive healthy children in the community. Since there has been no previous evidence of an association between any food type and DEC carriage, unhygienic handling, and preparation of matoke and pulses/legumes could be the reason for significant association with DEC carriage. Children 12–18 months old are more prone to DEC infections due to exploration and hand-to-mouth behavior. A detailed understanding is required on what proportion of positive cases developed severe symptomatology as well as fatal outcomes. The co-infection of pathotypes in the rapidly urbanizing environment needs to be investigated for hybrid or hetero-pathotype circulation that have been implicated in previous infection outbreaks.

**Funding:** The project Development of a comprehensive intervention to address foodborne enteric disease risks among young children living in low-income informal neighbourhoods of Maputo and Nairobi is supported by the Bill & Melinda Gates Foundation (BMGF) and the Foreign, Commonwealth and Development Office (FCDO) of the UK Government (INV-008449) and the CGIAR Research Program on Agriculture for Nutrition and Health, which is led by the International Food Policy Research Institute (IFPRI).

**Competing interests:** The authors have declared that no competing interests exist.

## Introduction

*Escherichia coli* is the predominant commensal gut bacteria [1, 2]. However, some *E. coli* strains carry virulence markers that enable them to cause infections in humans and other animals [3, 4]. As such, *E. coli* is included in the ESKAPE-E pathogens (*Enterococcus faecium*, *Staphylococcus aureus*, *Klebsiella pneumoniae*, *Acinetobacter baumannii*, *Pseudomonas aeruginosa*, *Enterobacter spp*., and *E. coli*). These pathogens are clinically important causing the majority of nosocomial infections and demonstrating increasing multi-drug resistance [5, 6]. Using virulence markers, *E. coli* colonize the gut of children and contribute to specific pathophysiology, causing gastroenteritis, invasive infections, urinary tract infections and other diseases [7]. These traits make diarrheagenic *E. coli* (DEC) antigenically distinct and have been used to classify them into six major pathotypes [8, 9]. These pathotypes include Shiga-toxin producing *E. coli* (STEC), enteropathogenic *E. coli* (EPEC), enterotoxigenic *E. coli* (ETEC), enteroaggregative *E. coli* (EAEC), entero-invasive *E. coli* (EIEC), diffusely adherent *E. coli* (DAEC) and the recent but rare adherent-invasive *E. coli* (AIEC) [10–13]. Enteropathogenic *E. coli* can be typical (tEPEC-carrying both *bfp* and *eae* genes) or atypical (aEPEC-carrying only *eae* gene). Typical EPEC attaches to intestinal cells via the *bfp gene*, thus associated with more severe diarrhea in young children than aEPEC that infects individuals of all age groups [14].

In low- and middle-income countries (LMICs), like high-income countries (HICs) *E. coli* and rotavirus are among the most important etiological agents of moderate-to-severe diarrhea [15]. Globally, 1.7 billion childhood diarrheal cases are reported annually, resulting in 525,000 deaths [16]. Diarrheagenic *E. coli* causes 80% of diarrheal illnesses in humans in Africa resulting in 70,000 deaths each year with children under the age of five being the most affected [16]. Diarrheal illness is a significant public health concern as it can lead to dehydration and other complications particularly in young children.

Risk factors associated with gut colonization and spread of DEC infection in humans include poor sanitation and hygiene practices such as open defecation, lack of clean water, and inadequate hand washing facilities [17]. Malnutrition and HIV/AIDS infections are also significant risk factors for DEC infection as they weaken the immune system and make individuals more susceptible to infections [18–21]. Overcrowded living conditions such as in refugee camps and slums can increase the risk of DEC infection due to the ease of person-to-person transmission [19]. Other risk factors include contact with infected individuals and animals, contaminated food and water (including street-vended foods and untreated water sources, such as rivers and lakes) [22].

As children transition from exclusive breastfeeding at 6–24 months to consumption of complementary foods and water, they are at increased risk of diarrheal disease [23, 24]. Recent studies conducted in Kenya on children presenting with diarrhea to various tertiary hospitals have reported DEC prevalence ranging from 34–60% [4, 25, 26]. In Egypt, DEC prevalence among children under five years of age with diarrhea in the community was 20.6% [12], while a study in Uganda reported a DEC prevalence of 38.2% among children with acute diarrhea [27]. However, there is lack of data on the prevalence of DEC pathotypes among asymptomatic weaning children in the community setting. The aim of this study was to estimate the detection prevalence of DEC pathotypes among asymptomatic children aged 6-24months in a peri-urban community of Nairobi, Kenya, and to identify risk factors for DEC detection.

## Methods

### Ethics consideration

Prior to commencing data collection, ethical approvals were obtained from the Research Ethics Committee of the London School of Hygiene and Tropical Medicine (Ref: 17188) and the Institutional Research Ethics Committee at the International Livestock Research Institute (Ref: ILRI-IREC2019-26). The ILRI-IREC is accredited by the National Commission for Science, Technology, and Innovation in Kenya (NACOSTI). In addition, project and individual student approvals were obtained from NACOSTI (License No: NACOSTI/P/21/10409). All study participants (i.e. adult caregivers) provided written informed consent before entry into the study.

### Study design, population and recruitment strategy

Our cross-sectional study was conducted in Dagoretti Sub-County, Nairobi between 1st May 2021 and 1st October 2021 as part of the Urban Infant Foodscape Project (UIF). The UIF project aimed to estimate foodborne disease (FBD) burden in young children in a low-income, high density urban environment by quantifying microbial contamination of pediatric foods and evaluating the risk factors for contamination and exposure across three domains: household, market and the production/supply chain and design intervention. Dagoretti South sub-county is a peri-urban area covering 29.1km$^2$ and comprises 9.87% (434,208/4,397,073) of Nairobi County's population [28]. It is a low-income peri-urban settlement characterized by high infectious disease burden, inadequate WASH (water, sanitation and hygiene) services and high human population density [29]. Dagoretti South sub-county was further stratified into the five different administrative areas (wards) and two wards, Uthiru/Ruthimitu and Riruta, were selected based on their proximity to health facilities (Fig 1). A complete list of all active community health volunteers (CHVs) from the two wards was obtained from the sub-county health coordinator. Out of this list, 100 CHVs were randomly selected using random number generator. However, 90 of the 100 CHVs were available to take part in the study. The 90 CHVs provided complete lists of eligible households in their catchment areas that had at least one child 6–24 months old. The households were stratified and up to 7 households were randomly selected from each CHV, resulting in a total of 585 households recruited into the study.

### Data and sample collection

A study nurse was paired with a CHV and visited each of the recruited households. After obtaining informed consent from an adult caregiver, the nurse conducted a questionnaire survey. The structured questionnaire covered questions relating to risk factors including poor sanitation and hygienic practices, dietary habits, household size (area and number of people), age and sex of child, contact with infected animals and consumption of contaminated food and water. Water sources were classified based on the WHO/UNICEF Joint Monitoring Program (JMP) "ladder" approach for assessing access to safe water supply, sanitation and hygiene services [30].

After the visit, the nurse left a set of diapers with the caregiver instructing the caregiver to remove the diaper from the child upon defecation without touching the inside, wrap and place it into a Ziplock bag labelled with household identification number and date. The diaper was then collected within 2 hours and transported in cold chain (cool box at 4˚C) to the International Livestock Research Institute (ILRI) laboratory for processing within 6 hours of collection.

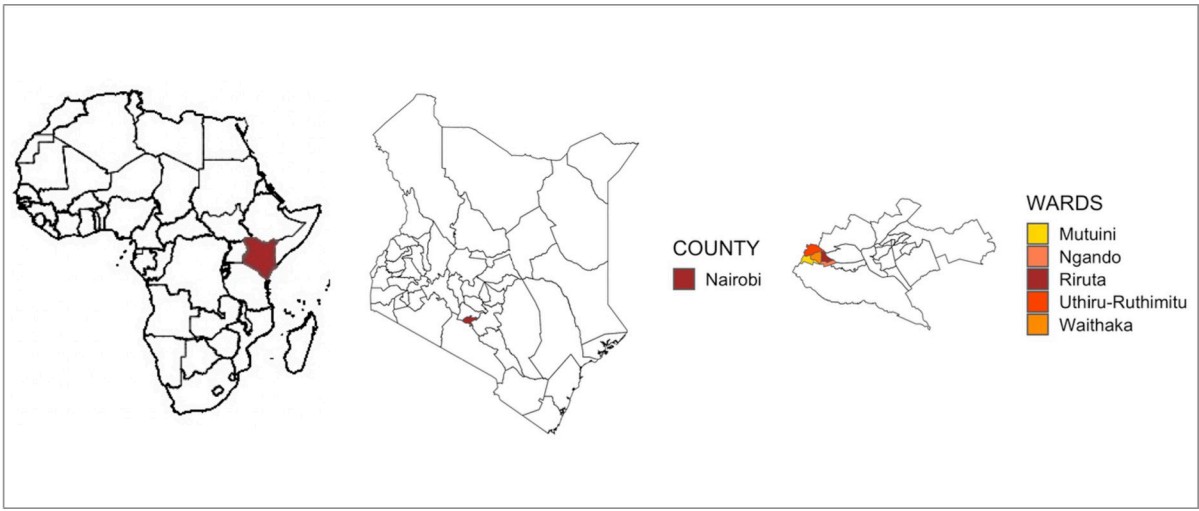

**Fig 1. Map of the study area indicating the wards of Dagoretti South sub-county within Nairobi County in Kenya.** Base layers were provided by the Humanitarian Data Exchange (https://data.humdata.org/) and ArcGIS Hub https://hub.arcgis.com/datasets.

## Isolation and identification of *E. coli* pathotypes

To isolate *E. coli*, stool samples were directly plated on differential media, MacConkey agar and incubated at 37˚C for 18-24h. Up to three lactose fermenting typical colonies appearing red/pink were biochemically tested by stabbing motility-indole-ornithine (MIO) media and streaking a trypticase soy agar (TSA) plate for isolation of pure culture. The TSA plates and MIO tubes were then incubated at 37˚C for 18-24h. After incubation, 2–3 drops of Kovacs reagent was added and all motility, indole and ornithine positive colonies considered presumptive *E. coli.* All the presumptive *E. coli* colonies were confirmed by Matrix Assisted Laser Desorption Ionization time of flight, mass spectrophotometry (MALDI-TOF MS), identifying them based on their protein profile (Fig 2).

To identify DEC pathotypes (ETEC, EPEC, EAEC, EIEC, and DAEC), specific virulence markers were used to pathotype the confirmed *E. coli* isolates using multiplex PCR. Typical EPEC was defined by presence of *bfp* (bundle-forming pilus) and *eae* (intimin) and absence of *stx* (Shiga-toxin) while atypical EPEC was characterized by presence of *eae* and absence of *bfp* and *stx* virulence genetic markers. Presence of *elt* (heat-labile toxin-LT) and/or *stl* (heat-stable-toxin-ST) was used to characterize ETEC, while *ipaH* (invasion plasmid antigen H) and *virF* (transcriptional regulator) were used for EIEC screening. Dr-binding adhesin family protein (*daaE*) was characteristic of DAEC while *aafII* (aggregative adherence fimbria II protein AafB) and *pic* (Serine protease autotransporter) were characteristic of EAEC presence [10, 13, 31]. The PCR conditions and primers are available in S1 File.

For Shiga toxin producing *E. coli* (STEC) seropathotype A (O157:H7) and seropathotype B (non-O157:H7) isolation, stool samples were plated onto Sorbitol MacConkey agar supplemented with cefixime tellurite (CT-SMAC) and incubated at 37˚C for 18-24h. Non-sorbitol fermenting colonies (NSFC) transparent or colorless with pale brownish appearance on CT-SMAC are typical of *E. coli* O157:H7. These NSFC were tested for immunochemical reaction with O157:H7 latex reagent (O157 antibody coated latex and control latex) according to manufacturer's instructions [32–34]. Isolates seropositive for O157 and ± H7 flagellar antigen were reported as presumptively positive for *E. coli* O157:H7 since several species cross-react with O157 antiserum. These presumptive colonies were confirmed as *E. coli* using MALDI

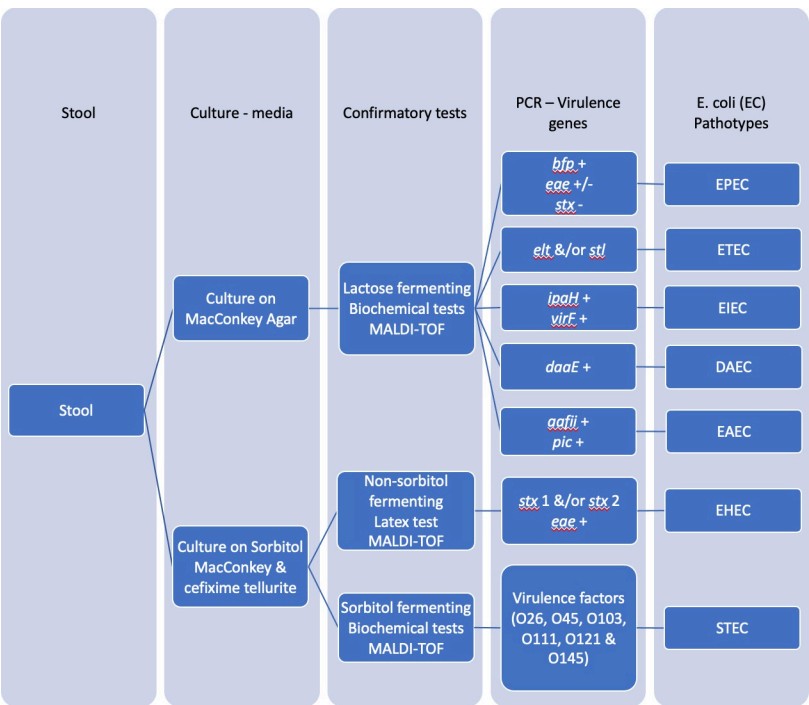

**Fig 2. Flow chart indicating the methodological approach to identifying different diarrheagenic *E. coli* in children's stool.** MALDI-TOF—Matrix Assisted Laser Desorption Ionization time of flight mass spectrophotometry; STEC—Shiga-toxin producing *E. coli*; EPEC—enteropathogenic *E. coli*; ETEC—enterotoxigenic *E. coli*; EAEC— enteroaggregative *E. coli*; EIEC—enteroinvasive *E. coli*; DAEC—diffusely adherent *E. coli*; AIEC—adherent-invasive *E. coli*.

TOF MS. PCR confirmation of their carriage of virulence markers stx 1 and/or 2 and intimin (eae) gene was done. These were confirmed as enterohemorrhagic *E. coli* (EHEC).

Sorbitol fermenting colonies (SFC) with red or pink appearance on CT-SMAC were biochemically characterized using MIO media as presumptive *E. coli*. These colonies were confirmed as *E. coli* using MALDI TOF MS. These were further pathotyped in multiplex PCR reactions targeting virulence markers specific for each serogroup, to characterize them as non-O157 (non-EHEC) STEC (O26, O45, O103, O111, O121 and O145) [35]. The NSF *Escherichia coli* (ATCC 700728) was used as a positive control, while the sorbitol fermenter *E. coli* (ATCC 25922) was used as a negative control.

## Statistical analysis

Descriptive summary statistics were computed to describe the detection prevalence of the six *E. coli* pathotypes. Presence/absence of an *E. coli* pathotype in stool was the primary outcome. The *lme4* package in R (version 4.2.2) was used to make mixed effects logistic regression models to identify risk factors associated with *E. coli* pathotype(s) carriage in children. Predictor variables considered were household size (number of people residing within a household), livestock ownership, manure/animal feces presence in the compound (as a subset of households keeping livestock), visible sign of rodents, and those relating to food type and food handling practices. Model selection was done in a two-step statistical logistic regression using univariable and backward stepwise multivariable analysis. The CHV identity was used as a random effect to account for clustering in the model regarding risk factors for *E. coli* pathotypes in children. Using a Fisher's test to assess the level of association of predictors identified during

univariable analysis, collinearity effect between these predictors was evaluated. Additionally, during the selection process, risk factors that exhibited more than a two-fold change in the logistic regression coefficients were also examined. All predictor variables with p value ≤0.1 during univariable analysis were retained for assessment in the multivariable analysis. A backward stepwise approach was used in the multivariable analysis to select the best model fit. Odds ratios of various predictors of *E. coli* pathotype prevalence and 95% confidence interval were calculated and considered statistically significant if the p-value was < 0.05. To evaluate two-way interactions between predictor variables, a likelihood ratio test was conducted and interactions were deemed significant if the p ≤ 0.05. Generalized Variance Inflation Factors (GVIF) were calculated to check for multicollinearity among the predictor variables. GVIF > 4 were considered a problem and the variable removed from the model. Model validity was checked by comparing the residuals plot to the fitted values plot for each fixed effect to assess for clustering and outliers, and deviations from the empirical and expected quantile distribution were determined to be not significant (p-value > 0.05) using the *simulateResiduals* function from the *DHARMa* package [36]. A Q-Q plot was visualized to detect deviations from the expected distribution including goodness-of-fit tests for correct distribution, overdispersion and outliers.

## Results

### Demographics of the study population

Stool samples were collected from 540 (92.3%) of the recruited 585 children. The caregivers of the 45 children who did not provide stool samples either declined or were unable to do so. The children either failed to produce stool within three days of follow-up or produced stool at night in which case these could not be collected and transported to the laboratories within 6 hours of collection. Of the sampled children, 274 were males while 266 were females,

**Table 1. Prevalence of diarrheagenic *E. coli* isolated from 6–24 months old asymptomatic children in Dagoretti South subcounty, Nairobi.**

| *E. coli* Pathotype | No. of children (N = 540) | Prevalence (95% CI) |
|---|---|---|
| O157:H7 STEC | 12 | 2.2% (1.3–3.8%) |
| non-O157:H7 STEC | 10 | 1.9% (1.0–3.3%) |
| EPEC | 35 | 6.5% (4.7–8.9%) |
| ETEC | 17 | 3.1% (2.0–4.9%) |
| EAEC | 21 | 3.9% (2.5–5.8%) |
| EIEC | 0 | 0 |
| DAEC | 1 | 0.2% (0.0–1.0%) |
| **Co-infections** | | |
| EPEC/O157:H7/O111 | 1 | 0.2% (0.0–1.0%) |
| EPEC/ETEC/EAEC | 1 | 0.2% (0.0–1.0%) |
| EPEC/ETEC | 4 | 0.7% (0.3–1.9%) |
| EPEC/EAEC | 2 | 0.4% (0.1–1.3%) |
| EAEC/ETEC | 4 | 0.7% (0.3–1.9%) |
| EPEC/O157:H7 | 1 | 0.2% (0.0–1.0%) |
| EAEC/O103 | 2 | 0.4% (0.1–1.3%) |
| EAEC/O157:H7 | 1 | 0.2% (0.0–1.0%) |
| EPEC/O145 | 1 | 0.2% (90.0–1.0%) |
| **Total** | **113** | **20.9% (17.8–24.5%)** |

**Table 2. Results of univariable analysis for risk factors for diarrheagenic _E. coli_ carriage in asymptomatic children in peri urban Dagoretti South sub-county, Nairobi.**

| Predictor | Number of observations | Number positive (%) | OR (95%CI) | p-value |
|---|---|---|---|---|
| **Community Health Unit** | | | | |
| Riruta | 145 | 33 (22.8) | 1 | Reference |
| Ruthimitu | 222 | 42 (34.4) | 0.79 (0.44–1.43) | 0.437 |
| Uthiru | 173 | 38 (22.0) | 0.94 (0.51–1.75) | 0.855 |
| **Infant Gender** | | | | |
| Female | 266 | 52 (19.5) | 1 | Reference |
| Male | 274 | 61 (22.3) | 1.21 (0.79–1.87) | 0.379 |
| **Age groups** | | | | |
| Age group (<12 months) | 170 | 31 (18.2) | 1 | Reference |
| Age group (12–18 months) | 188 | 50 (26.6) | 1.88 (1.11 – 3.19) | **0.02** |
| Age group (>18 months) | 182 | 32 (17.6) | 1.02 (0.58 – 1.80) | 0.94 |
| **Fever in the past one week** | | | | |
| No | 422 | 85 (20.1) | 1 | Reference |
| Yes | 117 | 28 (23.9) | 1.29 (0.77–2.16) | 0.33 |
| **Ever breastfed** | | | | |
| No | 7 | 3 (42.9) | 1 | Reference |
| Yes | 533 | 110 (20.6) | 0.32 (0.06–1.63) | 0.17 |
| **Type of hand washing facility** | | | | |
| Fixed handwashing facility inside the house | 26 | 5 (19.2) | 1 | Reference |
| Fixed handwashing facility in yard or compound | 52 | 11 (21.2) | 1.19 (0.35–4.07) | 0.78 |
| Mobile handwashing object such as a jug/bucket/kettle | 169 | 45 (26.6) | 1.72 (0.58–5.15) | 0.33 |
| No handwashing place (in the house, yard or compound) | 293 | 52 (17.7) | 1.01 (0.34–2.96) | 0.99 |
| **Main source of drinking water (JMP definition)** | | | | |
| Basic | 475 | 95 (20.0) | 1 | Reference |
| Safe | 65 | 18 (27.7) | 1.55 (0.83–2.90) | 0.17 |
| **Use utensils or hands to feed the infant** | | | | |
| Hands | 32 | 4 (12.5) | 1 | Reference |
| Utensils (e.g. spoon, fork) | 232 | 47 (20.3) | 2.05 (0.65–6.44) | 0.22 |
| Hands and utensils | 276 | 62 (22.5) | 2.35 (0.75–7.32) | 0.14 |
| **Liquid type consumed (broth)** | | | | |
| No | 457 | 92 (20.1) | 1 | Reference |
| Yes | 83 | 21 (25.3) | 1.43 (0.77–2.65) | 0.26 |
| **Food type consumed (pulses/legumes)** | | | | |
| No | 433 | 84 (19.4) | 1 | Reference |
| Yes | 107 | 29 (27.1) | 1.57 (0.94–2.64) | 0.09 |
| **Food type consumed (matoke-mashed bananas)** | | | | |
| No | 370 | 62 (16.8) | 1 | Reference |
| Yes | 170 | 51 (30.0) | 2.20 (1.40–3.46) | **0.001** |
| **Food type consumed (broth)** | | | | |
| No | 470 | 92 (19.6) | 1 | Reference |
| Yes | 70 | 21 (30.0) | 1.90 (1.01–3.56) | **0.05** |
| **Food reheating** | | | | |
| Boiled | 114 | 17 (14.9) | 1 | Reference |
| Warmed | 426 | 96 (22.5) | 1.66 (0.92–2.98) | 0.09 |
| **Milk reheated** | | | | |
| Boiled | 177 | 29 (16.4) | 1 | Reference |

_(Continued)_

**Table 2.** (Continued)

| Predictor | Number of observations | Number positive (%) | OR (95%CI) | p-value |
|---|---|---|---|---|
| Warmed | 363 | 84 (23.1) | 1.49 (0.91–2.43) | 0.11 |
| **Neighbor owns chickens** | | | | |
| No | 280 | 53 (18.9) | 1 | Reference |
| Yes | 260 | 60 (23.1) | 1.33 (0.86–2.06) | 0.20 |
| **Neighbor owns cat** | | | | |
| No | 255 | 48 (18.8) | 1 | Reference |
| Yes | 285 | 65 (22.8) | 1.34 (0.86–2.10) | 0.20 |
| **Sanitation facility (JMP)** | | | | |
| Unimproved | 73 | 16 | 1 | Reference |
| Limited | 105 | 26 | 1.11 (0.52–2.38) | 0.79 |
| Basic | 322 | 57 | 0.73 (0.37–1.43) | 0.36 |
| Safely managed | 40 | 14 | 1.84 (0.73–4.66) | 0.2 |

belonging to different age groups (<12 months, n = 170; 12–18 months, n = 188; and >18 months, n = 182) (Table 2).

## Diarrheagenic *E. coli* prevalence

Of the 540 samples, 503 (93.1%, 95%CI: 90.8–95.0%) tested positive for *E. coli*. According to the pathotyping results, 113 samples tested positive for diarrheagenic *E. coli*, with the prevalence of DEC in children 6–24 months being 20.9% (95%CI: 17.6–24.6%). From the 113 positive samples, 17 samples had co-infections (15 samples positive for two of the tested DEC pathotypes and two samples positive for three). This translated to a total of 132 isolates of diarrheagenic *E. coli* (Table 1). Enteropathogenic *E. coli* (n = 45, 34.1%) was the most prevalent pathotype (6 isolates tEPEC and 39 aEPEC), followed by EAEC (n = 31, 23.5%) and STEC (n = 29, 22.0%) among the positive samples.

## Risk factor analysis for diarrheagenic *E. coli* carriage in children

Table 2 presents the complete univariable analysis results for risk factors associated with DEC carriage in the 6-24m old children. The univariable analysis revealed that the odds of the outcome variable (carriage of DEC) were significantly associated with three predictor variables: age group 12-18m (OR = 1.88, 95%CI: 1.11 – 3.19, p = 0.02), consumption of matoke (mashed bananas) (OR = 2.2, 95%CI: 1.40 –3.46, p = 0.001), and consumption of broth (OR = 1.90, 95% CI: 1.01– 3.56, p = 0.05).

However, some variables lowered the odds of DEC carriage in children. The variables included: diarrhea in the past week (OR = 0.76, 95%CI: 0.75–0.76, p < 0.001) and food type biscuits (OR = 0.21, 95%CI: 0.06 – 0.74, p = 0.01) (S1 Table). There was no association between DEC carriage in children and number of people in the household, animal feces on premises, garbage presence on premises, number of households sharing toilets, visible signs of rodents, animal contact and animal ownership (S2 Table).

## Multivariable analysis

Table 3 summarizes results of the multivariable mixed effect logistic regression model for carriage of diarrheagenic *E. coli* in 6-24months old children. This best fit model had the lowest AIC of 541.0 and included: age group, food type (matoke- mashed bananas), food type (pulses/legumes), reheating food (warm enough to eat). The CHV identity was included in the

**Table 3. Multivariable analysis for carriage of diarrheagenic *E. coli* in asymptomatic children in Dagoretti sub-county, Nairobi.**

| | Pathogenic *E. coli* | | |
|---|---|---|---|
| *Predictors* | *Odds Ratios* | *95%CI* | *p value* |
| Age group [<12 months] | 1 | Reference | Reference |
| Age group [12–18 months] | 1.78 | 1.03 – 3.07 | **0.040** |
| Age group [>18 months] | 1.05 | 0.58 – 1.89 | 0.872 |
| Food type consumed—matoke (mashed bananas) [Yes] | 2.32 | 1.44 – 3.73 | **0.001** |
| Food type consumed–pulses/legumes [Yes] | 1.74 | 1.01 – 2.99 | **0.046** |
| Food reheated [Warm] | 1.71 | 0.93 – 3.14 | 0.082 |

model as a random effect which exhibited an interclass correlation coefficient (ICC) of 0.1. Risk factors associated with DEC carriage were: age group [12–18 months] (OR 1.78; 95%CI 1.03–3.07), feeding the infant with "matoke(mashed bananas)" (OR 2.32; 95%CI 1.44–3.73), and feeding infant with pulses/legumes (OR 1.74; 95%CI 1.01–2.99). During the stepwise backward variable selection process, all variables were found to have GVIF values below the threshold (GVIF<4), indicating no evidence of multicollinearity among the predictor variables. The final model's conditional R-squared value indicated that the risk factors "feeding the child with matoke", "feeding the child with pulses/legumes", and "age group of 12–18 months" explained 17.2% of the variation in the data.

The final multivariable mixed effects logistic regression model is presented in Fig 3 and the stepwise backward selection followed for model fitting is presented in S3 Table. During model validation, no significant clustering patterns of simulated residuals were observed. Further, overdispersion, zero-inflations and outliers tests did not yield any statistically significant results (p > 0.05).

## Discussion

In this study, we investigate the prevalence of pathogenic *E. coli* among asymptomatic children aged 6–24 months living in a peri-urban community with limited resources and examine any potential risk factors associated with the infection. The results provide insight on the potential food types and food handling practices that are significantly associated with DEC carriage in this vulnerable age group. Our findings of 20.9% prevalence is consistent with reports from clinical isolates in Africa and Asia: 21.2–41% in Kenya, 25% in Ethiopia (a review covering 10 articles investigating clinical DEC isolates from under 5 years old), 17–20.6% in Egypt (both healthy adults and children under 5 years of age with diarrhea) and 21% in India with 13 days—85 year old participants [1, 8, 12, 37–39].

Enteropathogenic *E. coli* was the most prevalent pathotype followed by EAEC and STEC. Typical EPEC is more commonly associated with diarrheal disease in infants and young children in developing countries [14]. Since these were presumptive healthy children, this could partly explain the low prevalence of tEPEC (4.6%), which was consistent with the Global Enteric Multicenter Study (GEMS) Kenyan tEPEC results of 3.5–5.2% among 0–23 months-old infants [40]. However, higher prevalence of tEPEC than aEPEC was reported in clinical isolates from Egyptian children [12]. EIEC was not isolated from any sample. Likewise, AIEC was not isolated and it is usually not associated with diarrhea. It is believed to be the etiological trigger of Crohn's disease, and so mostly isolated from Crohn's disease patients than from healthy individuals [11]. The same multicenter study (GEMS) reported ST-ETEC prevalence of 7% in 0–23 months old which compares with 9% reported in this study [15].

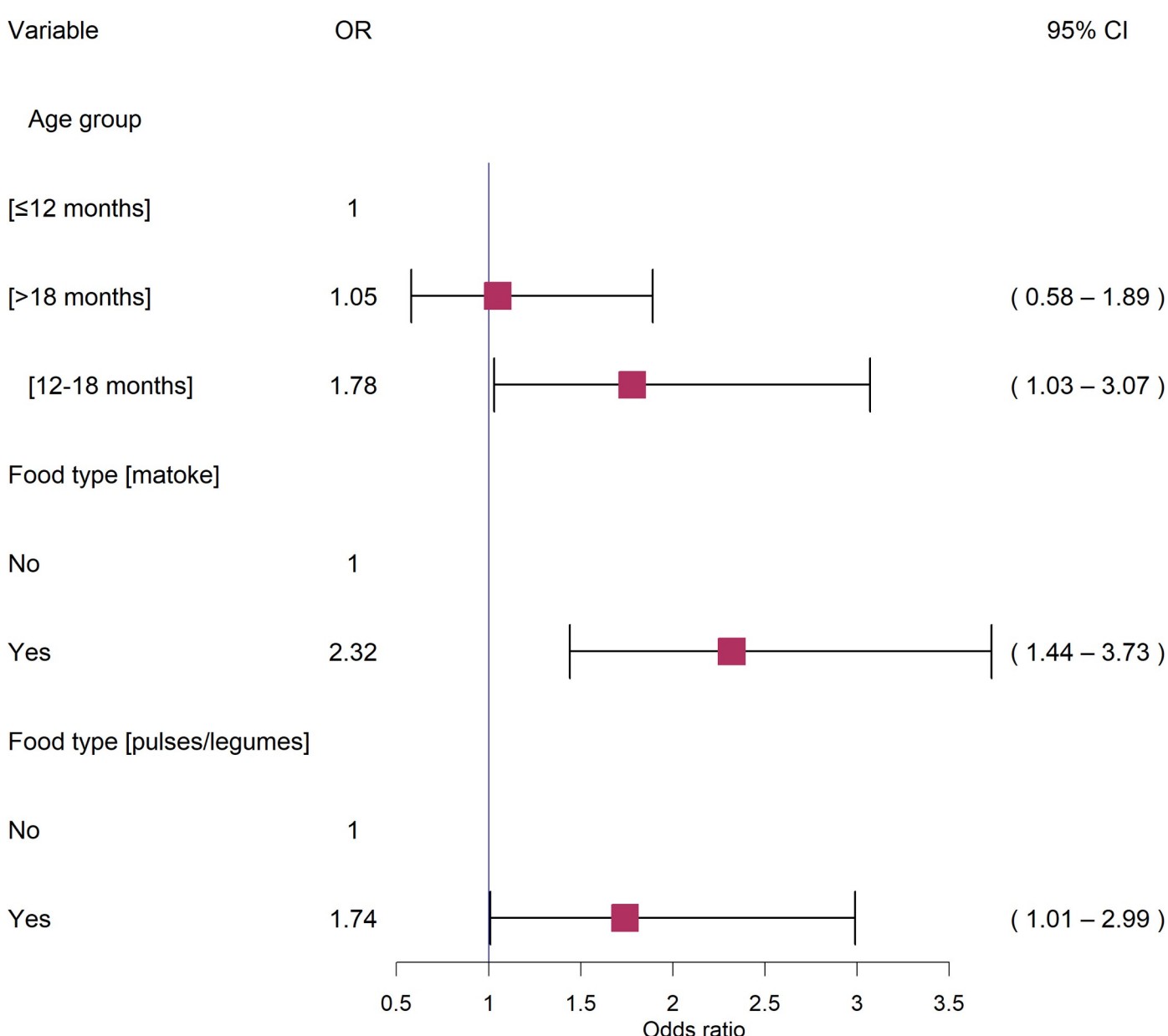

**Fig 3. Forest plot showing the final multivariable logistic model for diarrheagenic *E. coli* carriage in children.** For each category, the variable, levels, odds ratios (OR) with 95%CI is provided. Horizontal black lines represent 95%CI, while red boxes represent Odds ratios of each variable.

Diarrheagenic *E. coli* pathotype co-infections were observed in approximately 15% of the positive samples. This could be due to presence of virulence marker gene combinations like Shiga-toxin genes (*stx1* and 2) and intimin gene (*eae*) seen in STEC and EPEC. These have been used to define hybrid or hetero-pathogenic *E. coli* (like EPEC/STEC Hetero-Pathogen) that causes severe diseases [13]. This example of *E. coli* genomic plasticity was evident in the 2011 German outbreak that involved two pathotypes (EAEC/STEC), leading to recent interest in their significance as more virulent pathogens. Enterohemorrhagic *E. coli* has been linked to global outbreaks, causing severe symptomatology and even fatal outcomes in human hosts

[41], therefore its isolation from asymptomatic community children is of both epidemiological and public health concern.

Variables associated with DEC carriage in children at the multivariable model level were: age group 12–18 months, feeding infants with matoke (mashed bananas) and pulses/legumes. Grains and mashed bananas represent common staples in the Kenyan diet, while pulses/legumes and matoke were the predominant food categories investigated in this study. These foods are easily obtained from street vendors, and their use in households may be attributed to affordability, accessibility, nutritional composition, ease of preparation, and serving, making them popular choices for pediatric nutrition within the peri-urban settlement. Although no evidence has been provided linking any food type as a risk factor for DEC pathotype carriage in humans, previous studies have indicated that poor personal hygiene practices such as not washing hands before eating or after visiting the toilet, and unhygienic food handling and preparation practices from farm to fork can lead to contamination that eventually impact infant health [21, 22, 42, 43]. Children 12–18 months old are more mobile and curious and they explore their environment by putting things in their mouths and this can increase their exposure to contaminants [44–46]. This stage is also characterized with introduction of more solid foods and declining maternal antibodies [47, 48]. At the univariable analysis level, reheating food and milk to a temperature warm enough to eat was associated with DEC carriage. Optimal culture condition for *E. coli* include incubation at 37˚C, thus warming food (and not reheating to boiling) before feeding the infant could be providing an optimal growth condition for DEC. These hygienic practices are likely to be influenced by the household socioeconomic status, therefore recommended intervention strategies should be context specific [22, 49–51].

When participants who had diarrhea in the past week were evaluated for DEC positivity, significantly lower odds were observed. This could have been due to fecal shedding that results in lower pathogen positivity as diarrhea subsides and microbiota-mediated pathogen clearance [52]. There was no relationship between DEC carriage and number of household members which has been identified in previous studies as a risk factor together with animal feces/manure within the premises [53, 54]. Livestock keeping is a common practice in this rapidly urbanizing environment, and interactions between children and livestock is possible even with non-livestock keeping households [29, 54, 55]. However, there was consistence that livestock ownership had no association with DEC carriage. Further, given the complexity of DEC movement through the different One Health compartments, it is possible that DEC pathotypes move between humans and livestock via polluted environments and/or contaminated foods. This was evident by the significant association of matoke and pulses/legumes with DEC carriage. In a rapidly urbanizing environment like low-resource Dagoretti subcounty in Nairobi, human habitation, livestock keeping and the food supply chain are closely interconnected and indirect rather than direct DEC transmission events should be considered. This phenomenon therefore calls for a sustainable One Health intervention design targeting reduction in zoonotic disease carriage and transmission, and hygienic food handling and preparation practices along the value chain.

## Limitations

Using a PCR-based virulence gene profiling approach for pathotyping *E. coli* offers valuable insights into the pathogenicity potential of the isolates circulating in the community. However, this pathotyping approach has limited gene coverage, passible false negatives, and cannot determine functional expression. Further, with this approach, we were not able to capture emerging virulence factors thus hindering comprehensive characterization of our *E. coli* isolates. The decision to use this approach was based on cost, excluding more robust whole

genome characterization methods, which should be considered for future pathotyping research.

## Conclusion

The results of this study indicate a high prevalence of DEC pathotype carriage in asymptomatic children in the low-resource, densely populated, peri-urban neighborhood of Dagoretti sub-county-Nairobi, that is consistent with the numbers reported in clinical isolates both in Kenya and other countries. There is also evidence of mixed infections that have been shown previously to be indicative of hybrid and hetero-pathotypic infections, implicated in past outbreaks with severe disease symptomatology. This calls for further investigation to elucidate the genomic variation of *E. coli* for better understanding of the epidemiology of these infections. At the multivariable model level, two food types: matoke and pulses/legumes were significantly associated with DEC pathotypes carriage. This may suggest that the handling and preparation practices of these foods could be the cause. Further unidentified risks could be associated with this high carriage of DEC pathotypes in this vulnerable age group. There is no data presently on severe symptoms of gastritis and associated mortality from these asymptomatic DEC pathotype positives. Future research should consider following up asymptomatic community study participants to understand clinical outcomes and risks.

## Supporting information

**S1 File. PCR conditions for *Escherichia coli* pathotyping.**
(DOCX)

**S2 File. Inclusivity in global research questionnaire.**
(DOCX)

**S1 Table. Variables lowering odds of DEC carriage in children.**
(DOCX)

**S2 Table. Variables with no association with DEC carriage in children.**
(DOCX)

**S3 Table. Comparison of mixed-effects logistic regression models for risk factors for diarrheagenic *E. coli* carriage in 6–24 months old children.**
(DOCX)

## Acknowledgments

The authors wish to thank the participants for their time, the county officials for their support, community health workers and study nurses. Particularly, the authors would like to thank Catherine Mugo, Esther Gichache, Charity Sayo, Nicholas Ireri, John Ng'ang'a, Collins Mbuvi, Jean Amour, Mark Irumva, Immaculate Nyakabi, Hellen Omina and Cynthia Mukanda who helped with sample collection. Special thanks to Michael Ominde of Central Core Unit, ILRI.

## Author Contributions

**Conceptualization:** Noah O. Okumu, Julie Watson, Oliver Cumming, Elizabeth A. J. Cook.

**Data curation:** Noah O. Okumu, Elizabeth A. J. Cook.

**Formal analysis:** Noah O. Okumu, Dishon M. Muloi, Maud Akissi Amon-Tanoh, Elizabeth A. J. Cook.

**Funding acquisition:** Arshnee Moodley, Oliver Cumming.

**Investigation:** Noah O. Okumu, Linnet Ochien'g, Christine Mutisya, Alice Kiarie, Joseph O. Wasonga.

**Methodology:** Noah O. Okumu, Julie Watson, Oliver Cumming, Elizabeth A. J. Cook.

**Supervision:** Joseph J. N. Ngeranwa, Arshnee Moodley, Oliver Cumming, Elizabeth A. J. Cook.

**Validation:** Linnet Ochien'g.

**Writing – original draft:** Noah O. Okumu.

**Writing – review & editing:** Joseph J. N. Ngeranwa, Dishon M. Muloi, Arshnee Moodley, Julie Watson, Oliver Cumming, Elizabeth A. J. Cook.

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
