## [Decision Letter · Decision Letter 0]

1 Sep 2023

PGPH-D-23-01426

Risk factors for diarrheagenic Escherichia coli infection in children aged 6-24 months in peri-urban community, Nairobi, Kenya

Dear Dr. Okumu,

Thank you for submitting your manuscript to PLOS Global Public Health. After careful consideration, we feel that it has merit but does not fully meet PLOS Global Public Health’s publication criteria as it currently stands. Therefore, we invite you to submit a revised version of the manuscript that addresses the points raised during the review process.

We look forward to receiving your revised manuscript.

Kind regards,

Tinsae Alemayehu, MD

Guest Editor

Journal Requirements:

2. Please include a complete copy of PLOS’ questionnaire on inclusivity in global research in your revised manuscript. Our policy for research in this area aims to improve transparency in the reporting of research performed outside of researchers’ own country or community. The policy applies to researchers who have travelled to a different country to conduct research, research with Indigenous populations or their lands, and research on cultural artefacts. The questionnaire can also be requested at the journal’s discretion for any other submissions, even if these conditions are not met.  Please find more information on the policy and a link to download a blank copy of the questionnaire here: https://journals.plos.org/globalpublichealth/s/best-practices-in-research-reporting. Please upload a completed version of your questionnaire as Supporting Information when you resubmit your manuscript.

3. Some material included in your submission may be copyrighted. According to PLOS’s copyright policy, authors who use figures or other material (e.g., graphics, clipart, maps) from another author or copyright holder must demonstrate or obtain permission to publish this material under the Creative Commons Attribution 4.0 International (CC BY 4.0) License used by PLOS journals. Please closely review the details of PLOS’s copyright requirements here: PLOS Licenses and Copyright. If you need to request permissions from a copyright holder, you may use PLOS's Copyright Content Permission form.

Potential Copyright Issues:

Fig 1&2: please (a) provide a direct link to the base layer of the map (i.e., the country or region border shape) and ensure this is also included in the figure legend; and (b) provide a link to the terms of use / license information for the base layer image or shapefile. We cannot publish proprietary or copyrighted maps (e.g. Google Maps, Mapquest) and the terms of use for your map base layer must be compatible with our CC-BY 4.0 license. 

"

Additional Editor Comments (if provided):

Reviewers' comments:

Reviewer's Responses to Questions

**Comments to the Author**

1. Does this manuscript meet PLOS Global Public Health’s publication criteria? Is the manuscript technically sound, and do the data support the conclusions? The manuscript must describe methodologically and ethically rigorous research with conclusions that are appropriately drawn based on the data presented.

Reviewer #1: Yes

Reviewer #2: Yes

Reviewer #3: Yes

Reviewer #4: Yes

2. Has the statistical analysis been performed appropriately and rigorously?

Reviewer #1: Yes

Reviewer #2: Yes

Reviewer #3: I don't know

Reviewer #4: Yes

3. Have the authors made all data underlying the findings in their manuscript fully available (please refer to the Data Availability Statement at the start of the manuscript PDF file)?

Reviewer #1: Yes

Reviewer #2: Yes

Reviewer #3: Yes

Reviewer #4: Yes

4. Is the manuscript presented in an intelligible fashion and written in standard English?

Reviewer #1: Yes

Reviewer #2: Yes

Reviewer #3: Yes

Reviewer #4: Yes

5. Review Comments to the Author

Reviewer #1: The manuscript entitled "Risk factors for diarrheagenic Escherichia coli infection in children aged 6-24 months in peri-urban community, Nairobi, Kenya" shows interesting findings of the prevalence and risks associated with infectionthast could cause diarrhoea in children.

The methodology and statistical analysis is appropriate.

Questions to the authors:

- Several potential risk factors were thought to contribute to infection, was climate associated conditions also considered eg. an increase in infection in the rainy season compared to the dry season?

- Are environmental samples planned to be taken

- Are there any interventions that are in place regarding safe food preparation practises in the county?

- A follow-up study as to how of the children infected with E.coli actually went on to develop diarrhoeal disease, would be interesting as well.

Reviewer #2: The study is well described and reaches its goal. Though it's not the objective of the study it would have been great if the study was a cohort study and follow the child whether they develop diarrhea or not and which patient depending on the pathotype will have complications.

Reviewer #3: On page 5- would suggest explaining removal of diaper 'aseptically'- as this surely is not possible in the community.

In results section on page 5, could you be more specific about why caregivers were 'unable to do so' (provide a sample)- was this because the child was unwell or had loose stools?

Does Table 1 have p-values?

Should there be a demographics table before Table 1- which describes the sampled populations?

Apologies I find Table 4 difficult to interpret- if just formulas for the regression model, perhaps more appropriate for an Appendix section.

Reviewer #4: I appreciate the opportunity to review PGPH-D-23-01426 for PLOS Global Public Health. Noah Okoth Okumu and colleagues analyzed the risk factors for diarrheagenic Escherichia coli (DEC) infection in young children in a peri-urban community in Nairobi, Kenya. Considering the burden of E. coli during early childhood in LMICs, identifying a significant prevalence of pathogenic DEC circulating in the community among healthy children and the risk factors associated with carriage of DEC pathotypes is extremely valuable to other African countries with rapidly urbanizing environments. Therefore, I believe this report is original and relevant to the field of Global Public Health. Please find my considerations to improve this manuscript for publication in the PDF file annexed.

6. PLOS authors have the option to publish the peer review history of their article (what does this mean?). If published, this will include your full peer review and any attached files.

**Do you want your identity to be public for this peer review?** For information about this choice, including consent withdrawal, please see our Privacy Policy.

Reviewer #1: No

Reviewer #2: **Yes: **Elham Sany, MD, Pediatrician, Infectious disease fellow.

Reviewer #3: No

Reviewer #4: **Yes: **Daniel Jarovsky

---

## [Editor Report · Decision Letter 1]

18 Oct 2023

Risk factors for diarrheagenic Escherichia coli infection in children aged 6-24 months in peri-urban community, Nairobi, Kenya

PGPH-D-23-01426R1

Dear Mr Okumu,

We are pleased to inform you that your manuscript 'Risk factors for diarrheagenic Escherichia coli infection in children aged 6-24 months in peri-urban community, Nairobi, Kenya' has been provisionally accepted for publication in PLOS Global Public Health.

Best regards,

Tinsae Alemayehu, MD

Guest Editor